# The Effect of Exogenous Melatonin on Eating Habits of Female Night Workers with Excessive Weight

**DOI:** 10.3390/nu14163420

**Published:** 2022-08-19

**Authors:** Luciana Fidalgo Ramos Nogueira, Cibele Aparecida Crispim, José Cipolla-Neto, Claudia Roberta de Castro Moreno, Elaine Cristina Marqueze

**Affiliations:** 1Department of Epidemiology, Public Health Graduate Program, Catholic University of Santos, Av. Conselheiro Nébias 300, Vila Mathias, Santos 11015-001, Brazil; 2Chrononutrition Research Group, Faculty of Medicine, Federal University of Uberlândia, Av. Pará, 1720, Bloco 2U, Uberlandia 38405-320, Brazil; 3Department of Physiology and Biophysics, Institute of Biomedical Sciences, University of São Paulo, Av. Lineu Prestes 1524, Cidade Universitária, Butantã, Sao Paulo 05508-000, Brazil; 4Department of Health, Life Cycles and Society, School of Public Health, University of São Paulo, Av. Dr. Arnaldo 715, Cerqueira César, Sao Paulo 01246-904, Brazil; 5Department of Psychology, Stress Research Institute, Stockholm University, 16 Frescati Hagväg, 10405 Stockholm, Sweden

**Keywords:** melatonin, dietary supplements, eating behavior, circadian dysregulation, night work

## Abstract

Background and Aims: Melatonin is a pineal hormone that plays an important role as an endogenous synchronizer of circadian rhythms and energy metabolism. As this circadian component has been closely related to eating behavior, an important question on this topic would be whether melatonin administration could influence eating habits. However, this topic has been rarely studied in the literature in individuals with excessive weight and chronic circadian misalignment, such as shift workers. Therefore, the present study aims to evaluate the effects of exogenous melatonin administration on the quali/quantitative aspects and temporal distribution of food intake in female night workers with excessive weight (overweight and obesity). An additional aim is to evaluate the association of the referred outcomes with circadian misalignment and chronotype. Methods: A randomized, double-blind, placebo-controlled, crossover clinical trial was conducted with 27 female nursing professionals with excessive weight who worked permanent night shifts. The protocol was implemented under real-life conditions for 24 weeks, in two randomly allocated conditions (12 weeks of melatonin and 12 weeks of placebo). The quali/quantitative aspects of food intake (NOVA classification, total energy intake and the proportion of calories from macronutrients) and meal timing were assessed using food diaries. Timing for every meal recorded in the diaries was assessed to evaluate the temporal distribution of food intake. Generalized estimating equations were performed for each dependent variable. Results: No significant modifications in total energy intake, macronutrient distribution, types of foods consumed, and meal timing were observed after melatonin administration. Different levels of circadian misalignment and chronotype did not interfere with these results. Conclusion: Eating habits of female night workers with excessive weight remained unchanged after melatonin administration, and no association of these results with circadian misalignment and chronotype was found. These results suggest that the metabolic effects of melatonin may occur independently of food intake.

## 1. Introduction

The modern world, including the growing percentage of workers involved in night shift work, challenges circadian and energy homeostasis by promoting food intake during the rest phase [1,2]. The desynchronization between the suprachiasmatic nucleus and the temporal clues of the peripheral tissues in shift workers has been associated with a higher perceived appetite for calorically dense food, later meal timing, and more snacks before sleep, independent of sleep duration [3,4]. These poorer eating habits have been linked with a higher prevalence of metabolic diseases in shift workers [5,6].

It is estimated that 20 to 30% of the economically active populations in North America, Canada, Europe, and Brazil are involved in shift work that includes night shifts [7,8,9,10,11]. Night work is considered the most extreme situation of circadian misalignment [12]. Previous findings suggest that the reduction in total daily energy expenditure during night shifts [13], as well as lower energy expenditure in response to dinner [14], may increase weight gain and the risk of obesity and adverse metabolic outcomes [13]. In addition, it has been suggested that evening-type individuals, for example, are used to eating at unfavorable eating times [15], showing high energy consumption late at night [16]. The same temporal distribution of food intake is observed in shift workers [17].

Food intake has been shown to be an important synchronizer for peripheral clocks [18] and, therefore, a modifiable temporal cue for the circadian system that can be influenced by innumerous factors [19]. One of these factors might be the plasma concentration of melatonin, a hormone produced by the pineal gland that plays an important role as an endogenous synchronizer of the circadian rhythms and energy metabolism, including energy intake and expenditure [20,21].

Until now, only three studies have evaluated the effects of exogenous melatonin on night workers in real-life conditions [22,23,24], but did not evaluate eating habits. These studies assessed the effect of melatonin on (1) circadian misalignment: administration prior to daytime sleep attenuated interference from circadian alerting process, demonstrating both phase-shifting and sleep-promoting actions [22]; (2) sleep and alertness: administration reduced sleep problems and increased alertness during working hours [23]; and (3) body weight: administration reduced body weight, BMI, and waist and hip circumferences [24].

Additionally, it is important to highlight that the participants of this study had excessive weight (overweight and obesity), and that body mass seems to affect daily rhythmicity and concentrations of circulating metabolites that express circadian rhythms [25]. In addition, it has been previously established that circadian rhythms and metabolism are closely linked, and that meal timing plays a role in synchronizing peripheral circadian rhythms in humans, which is particularly relevant for shift workers [26]. The choice of focusing on females is due to the higher prevalence of excessive weight in women [27], and because nursing is a female-dominated occupation [28].

Therefore, the present study aims to evaluate the effects of melatonin administration on the quali/quantitative aspects and temporal distribution of food intake. An additional aim is to evaluate the association of the referred with circadian misalignment and chronotype. The question we expect to answer is: Does melatonin administration influence food choices, the quali/quantitative aspects and temporal distribution of food intake among female night workers with excessive weight? Given the potential of melatonin to regulate energy metabolism and to synchronize circadian rhythms including energy metabolism, we hypothesize that melatonin administration will be able to improve eating habits.

## 2. Materials and Methods

### 2.1. Design and Sampling

A randomized, double-blind, placebo-controlled, crossover clinical trial was conducted with female nursing professionals (nursing technicians and nurses) with excessive weight who worked permanent night shifts in a large private hospital in São Paulo, SP, Brazil. The protocol was implemented under real-life conditions for 24 weeks. All nurses worked in a 12 × 36 scheme (12 h on night shifts, from 19:00 to 7:00 h, followed by 36 h off) and had one day off every 15 days.

The sample was calculated considering a test of difference of means of repeated measures (within–between interaction), the effect size of 0.30, alpha error of 5%, two groups (intervention and CPD), and two measures (melatonin and placebo), in which the sample of 27 people represented a power of 85% (G×Power). Detailed information on the study sample is published elsewhere [24].

The inclusion criteria were women aged 20 to 50 years, with body mass index (BMI) ≥25 and <40 kg/m^2^, working for at least six months in the current night shift system, who declared having no intention of following any restricted diets and starting new physical activities while participating in the study. The exclusion criteria were pregnant or lactating women; having children under one years old; climacteric or menopause; eating disorders diagnosed by a physician; having a second night’s work; regular use of medications or dietary supplements that influence sleep, alertness, and the circadian timing system; abusive use of drugs and alcohol; past and/or current history of psychiatric, neurological, circadian or sleep disorders diagnosed by a physician; metabolic disorders (except treated type 2 diabetes mellitus and dyslipidemia); cardiovascular diseases (except treated systemic arterial hypertension); chronic inflammation and/or infection diagnosed by a physician; anemia and/or having donated >400 mL of blood in the last three months preceding the study; major surgery in the six months preceding the study.

### 2.2. Study Protocol

Participants were recruited from March 2018 to June 2019. They were randomly allocated into two groups using codes generated by a computer. Since it is a crossover trial, in the first phase, one group received melatonin and the other group received a placebo; then, in the second phase, the participants switched groups. Identical tablets of 3 mg of fast-release melatonin or placebo (Aché Pharmaceutics^®^, Sao Paulo, Brazil) were orally administered for 12 weeks each. Participants were instructed to take one tablet, one hour before bedtime, exclusively when they slept at night, that is, in the nights between shifts and days off, and never during the day. It is important to highlight that fast-release melatonin was chosen due to its higher effectiveness in circadian synchronization in comparison to other types [29]. Since melatonin is completely excreted in urine within 24 h after administration [30], no washout period was performed between the two phases of the study. The average number of days of melatonin administration was 45 days (EP 10.3 days) and the use of placebo was 44.3 days (EP 8.2 days). More information about the study protocol is provided elsewhere [24], and Appendix A presents the study flow chart.

### 2.3. Study Variables

#### 2.3.1. Outcomes

Quantitative aspect of the diet: Total energy intake and macronutrient distribution

Food intake was assessed by food diaries, which participants were previously instructed to fill in on typical workdays and days off (from 19:00 to 19:00 h). They completed these diaries two days per month while participating in the study, therefore totaling seven months (baseline plus 24 weeks of intervention) and 14 days recorded (two days per month, being one workday and one day off).

The participants were also given instructions to provide as detailed information as possible about food and drinks consumed, including brand names, ingredients used in homemade recipes, and meal timing. Portion sizes were estimated using household measures and subsequently converted to mass units of measurement (g) and capacity (mL), according to Pinheiro et al. [31]. Due to cultural differences between Brazilian and North American eating habits, the composition of typical Brazilian preparations was manually added to the software database using the Brazilian Food Composition Table [32] and labels of industrialized products.

The diaries were reviewed by a nutritionist to obtain additional clarifications, when necessary, and analyzed using the Nutrition Data System for Research software (NDSR, 2007). The total energy intake (TEI) and the proportion of calories from each macronutrient (carbohydrate, protein, and fat) in relation to TEI were calculated.

Qualitative aspect of the diet: Food classification according to processing

Every item in the reviewed food diaries was categorized as unprocessed or minimally processed, processed, or ultra-processed foods according to the NOVA classification [33]. NOVA classifies food into four groups based on the nature, extent, and purpose of the industrial processing: (1) unprocessed or minimally processed foods, such as rice, beans, frozen meat, and milk; (2) processed culinary ingredients, such as vegetable oils and table sugar; (3) processed foods, such as vegetables in brine and cheeses; and (4) ultra-processed foods, such as carbonated soft drinks, meat products, and instant noodles. A detailed description of the NOVA classification can be found in [33].

All the foods consumed were coded as a number corresponding to a food group (*n* = 5.549) according to NOVA. When a direct classification was not possible, such as homemade preparations and foods containing items from different groups, as well as culinary ingredients, the classification of the main ingredient was considered [33].

Temporal distribution of food intake

Timing for every meal in the food diaries was grouped into four categories: 19:00–00:59 h, 01:00–06:59 h, 07:00–12:59 h and 13:00–18:59 h.

#### 2.3.2. Independent Variables

Composite Phase Deviations

The circadian misalignment was estimated using the composite phase deviation (CPD) [34] of actigraphy-based mid-sleep, which is calculated from the sleep onset and offset data. Participants wore actigraphs for 10 consecutive days (ActTrust and Basic Motionlogger Actigraph, Condor Instruments^®^, Sao Paulo, Brazil) and filled in sleep activity diaries to validate the recorded data. The CPD was calculated according to the following equation:$$\left|CPD_{i}\right|=\left|\frac{x_{i}}{y_{i}}\right|=\;\sqrt{x_{i}^{2}+y_{i}^{2}}$$
in which *CPD_i_* = composite phase deviation on day *i*; *x_i_* = distance of mid-sleep on day *i* to chronotype (MSFNsc)*; *y_i_* = distance of mid-sleep on day *i* to previous day *i* − 1. *MSFNsc = chronotype (mid-sleep on free days after night shifts, corrected for over-sleep).

Chronotype

Chronotype was assessed using the mid-point of sleep on free days after night shifts, corrected for over-sleep (MSFNsc) derived from the Munich Chronotype Questionnaire for shift work (MCTQshift) [35].

#### 2.3.3. Descriptive Variables

The participants went through a two-week baseline period in which a self-administered questionnaire was completed. Sociodemographic (age, marital status, education) and work characteristics (current position, reason to work at night, lifetime exposure to night work, second job) and health behaviors (alcohol intake, smoking, physical activity) were evaluated.

Body weight and height were assessed at baseline, and body weight was also assessed in the last 10–15 days of the first and second phases of the study. Both measurements were performed according to the standardization method by Lohman et al. [36]. Body weight was assessed to the nearest 0.1 kg using a calibrated digital balance. Height was assessed using a wall-mounted portable stadiometer to the nearest 0.1 cm. Body mass index (BMI) was calculated as the body weight (kg) divided by the height squared (m^2^).

The energy requirements of the participants were individually calculated by the equations of estimated energy requirements (EER) [37].

### 2.4. Statistical Analysis

Descriptive data are shown as mean ± standard error (SE). Shapiro–Wilk’s test was used to test data normality. Generalized estimating equations (GEE) with the Bonferroni post hoc test were used to analyze the effect of melatonin and placebo administration on quali/quantitative aspects and temporal distribution of the diet. Seven models were performed for each dependent variable (TEI; percentages of daily calories taken from unprocessed or minimally processed, processed, ultra-processed foods; carbohydrate, protein, and fat; and meal timing).

The percentages of calories for each category of processing were obtained: (1) calculating the average intake of the workday and the day-off registered at the baseline and during the last four weeks of each phase of the study (weeks 12 and 24), and (2) calculating the average consumption from the 12 weeks of melatonin administration, and then from the 12 weeks of placebo. The decision to use the values for the end of each phase of the study is due to the absence of significant month-to-month differences in food intake in previously tested models.

Additionally, seven models were performed to analyze the effect of the interventions on the same dependent variables aforementioned in association with CPD, and another seven in association with chronotype. Both CPD and chronotype were treated as continuous variables. All models were adjusted for lifetime night work exposure. Gamma distribution with log link was chosen considering the smaller quasi-likelihood under the independence model criterion (QIC). Statistical analyses were performed using SPSS version 20.0 (SPSS Inc., Chicago, IL, USA). The significance level was set at 5%.

### 2.5. Ethical Aspects

Ethical issues regarding research involving human beings have been duly respected. The research project was approved both by the Ethics Committee of the School of Public Health of the University of São Paulo (process number 2.450.682) and the Ethics Committee of the hospital where it was conducted (process number 2.489.636). All participants gave written and informed consent to participate in the study. The clinical trial was registered at the Brazilian Registry of Clinical Trials-ReBEC (number RBR-6pncm9) and was developed according to the Consolidated Standards of Reporting Trials [38].

## 3. Results

Sociodemographics, work characteristics, health behaviors, BMI, CPD, and chronotype are shown in Table 1. All the participants were female, and the majority were nursing technicians, married, with a major degree. The main reason mentioned by the participants to work at night was to reconcile work with home and/or children’s care, and the majority did not have a second job. None were smokers, and most consumed alcohol only on special occasions.

The analysis of the effect of melatonin and placebo administration showed no significant results. No associations between the quali/quantitative aspects of the diet and CPD or chronotype were found as well (Table 2).

Figure 1 presents the estimated energy requirements and total caloric intake after melatonin supplementation and placebo.

The mean EER of the participants at baseline was 2357.3 kcal/day. The mean TEI after melatonin administration was 1421.1 (SE = 88.3) kcal/day, and the mean TEI after placebo administration was 1532.9 (SE = 83.8) kcal/day (GEE *p* = 0.15). Additionally, no statistically significant differences in TEI between the intervention phases in association with CPD (melatonin: 1469.0 ± 262.4 kcal/day, placebo: 1482.7 ± 183.0 kcal/day, GEE *p* = 0.94) nor chronotype (melatonin: 1418.6 ± 174.5 kcal/day, placebo: 1531.8 ± 138 kcal/day, GEE *p* = 0.99) were observed.

In regard to the temporal distribution of food intake, no differences in meal timing between the intervention phases were observed (Figure 2). No associations between the temporal distribution of food intake and CPD or chronotype were found as well, as shown in Table 2.

## 4. Discussion

The present study showed that total energy intake, macronutrients distribution, types of foods consumed, and meal timing remained unchanged after melatonin administration in female night workers with excessive weight. No associations between these outcomes and circadian misalignment or chronotype were found either. These results contradict our initial hypothesis that melatonin administration would be able to improve eating habits.

### 4.1. No Effects of Melatonin on Energy, Quali/Quantitative Intake and Meal Timing

Little is known about the effect of melatonin administration on food intake [39]. In the present study, we found that melatonin administration had no effect on quali/quantitative aspects of the diet in female night workers with excessive weight. One of the hypotheses to justify our findings was recently emphasized by a systematic review that evaluated the effects of melatonin supplementation on eating habits and appetite-regulating hormones [39]. This review stated that melatonin’s effects on energy metabolism can occur independently of food intake. Furthermore, the only clinical trial included in the review found no difference in food intake after 84 days of melatonin supplementation (6 mg/day) in healthy men [40]. Although our results in the present study confirm these findings from the literature so far, the need for more randomized clinical trials to confirm these findings is evident.

Previous studies have reported that hunger increases and satiety decreases after a night shift, and that food preferences and appetite are altered to high-calorie density foods [41,42], that is, ultra-processed products. These foods contributed ~35% of TEI in the present study, and no changes were observed after the intervention. This amount is notably higher than the 22.7% observed in a cross-sectional analysis from the Brazilian Longitudinal Study of Adult Health (ELSA-Brasil) cohort [43]. The higher intake might be related to chronic fatigue and sleep disturbances frequently experienced by shift workers, which promotes hedonic control of food intake and decreases the motivation to prepare meals, facilitating the choice of ready-to-eat, less healthy options [44].

The implication of modern industrial processing on health is overall underrated, but the quality of the energy consumed deserves particular attention in specific groups at higher risk of metabolic disorders due to melatonin suppression, such as night shift workers. The ELSA-Brasil cohort consumed ~65% of TEI from unprocessed or minimally processed foods [42], while the participants of the present study consumed ~50%. Even though melatonin supplementation did not change the contribution of this type of food to TEI, factors influencing eating behavior still need to be better understood. According to Gupta et al. [45], another interplay that still needs to be better understood is the role of obesity on food choices. Evidence suggests that consuming unprocessed or minimally processed foods regularly has a protective effect against metabolic disorders [46]. In this context, the low intake of this type of foods by specific groups, such as shift workers, deserves attention in future studies concerning diet.

Excessive weight is primarily caused by a positive energy balance, that is, a state in which energy intake is higher than its expenditure [47]. However, the female night workers with excessive weight participating in our study consumed an average of 900 kcal/day less than their EER. Previous studies have shown that insufficient sleep resulted in increased energy intake, while adequate sleep decreased energy intake [48,49]. Our hypothesis on this topic is that energy intake would be higher than energy requirements due to circadian misalignment, and that melatonin administration might be able to mitigate the adverse metabolic effects of night work [50,51]. However, this hypothesis was not confirmed either and more studies are needed.

### 4.2. No Effects of Melatonin Supplementation on Food Intake in Association with Circadian Misalignment or Chronotype

Our findings in the present study showed melatonin administration had no effect on quali/quantitative aspects of the diet in female night workers with excessive weight, independent of their chronotype or level of circadian misalignment. A recent clinical trial published by our research group showed that 12 weeks of melatonin administration reduced circadian misalignment by 21% in the same cohort of female excessive night workers from the present study [24]. These results from our previous study led us to believe that greater changes in eating behavior would be more relevant in people who reduced their level of circadian misalignment, but this did not occur in the present study. This reinforces the aforementioned information that melatonin’s effects on energy metabolism, which has been postulated in the literature [20], can occur independently of food intake [52].

Regarding an interplay between melatonin administration, chronotype, and food intake, we hypothesized that morning individuals would benefit the most from the intervention since this chronotype has been associated with high circadian misalignment [24]. However, this hypothesis was not confirmed. Moreover, our previous study [24] showed that the administration reduced the body weight of the early chronotypes—the ones that suffer all the adverse effects of night work the most [8]. In this context, it is reasonable to assume that these individuals would respond differently to the intervention in comparison to the vespertine ones, but this hypothesis was not confirmed either. More randomized clinical trials are needed to confirm our findings.

### 4.3. Limitations and Strengths

The present study has some limitations. The same dosage of melatonin was administered to all participants, but melatonin’s metabolism has interindividual variations. The response to artificial light exposure at night also varies, leading to lower or higher melatonin suppression according to individual sensitivity and light intensity [53]. Hence, pharmaceutical formulation and dosage of exogenous melatonin could be individually considered [54]. Another possible limitation is that, although the participants were instructed to record food intake on typical days, it is not possible to guarantee that the records truly reflect their dietary behaviors in everyday life. It is also important to mention that the participants had the option of a nutritionist-planned dinner provided by the hospital. In addition, all units had a pantry in which they could store food brought in from outside the hospital, as well as have meals. This particular reality can influence the results obtained with regard to eating habits. Lastly, assessments of appetite-regulating hormones, i.e., leptin and ghrelin, might contribute toward understanding the lack of effect of melatonin administration on meal timing. However, these assessments were not performed in this study.

On the other hand, there are some strengths to be highlighted. A significant advantage of the present clinical trial is that, even though studies on pharmacokinetics in humans are limited, exogenous melatonin has shown to be safe and lacks adverse effects in comparison to placebo [54]. Additionally, fewer studies have employed within-subject comparisons and collected data regarding workdays and days off [42,55,56]. Lastly, assessing the dietary habits of female night workers with excessive weight in real-life conditions is one of the main strengths of the study. The qualitative aspect of the diet, especially, has been generally neglected in experimental studies [33] and adds important knowledge on the eating behaviors of this specific population.

## 5. Conclusions

The present study showed that the eating habits of female night workers with excessive weight remained unchanged after 12 weeks of intermittent exogenous melatonin administration. Circadian misalignment and chronotype did not interfere with these results. These results suggest that the metabolic effects of melatonin may occur independently of total energy intake, macronutrient distribution, food processing level, and temporal distribution of food intake.

## Figures and Tables

**Figure 1 nutrients-14-03420-f001:**
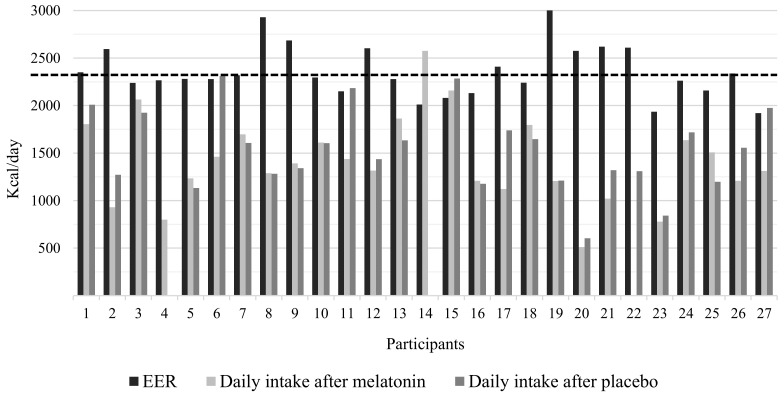
Estimated energy requirements and total caloric intake after melatonin and placebo administration. EER: estimated energy requirement. The dashed line represents the mean EER (kcal/day) of the participants. No food intake was recorded in the diaries after placebo administration for participants 4 and 14, and after melatonin administration for participant 22.

**Figure 2 nutrients-14-03420-f002:**
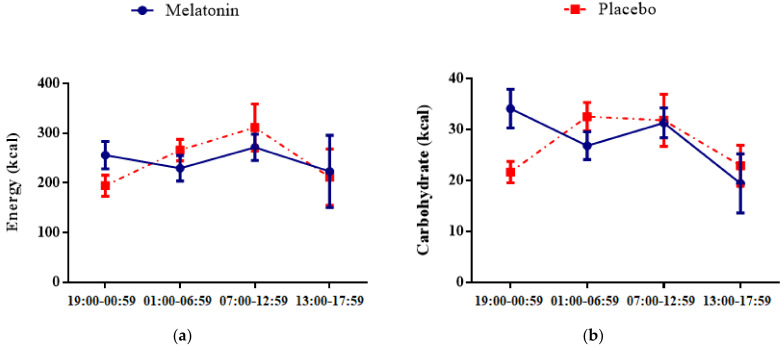
Temporal distribution of food intake after melatonin and placebo administration. Vertical lines represent standard error of mean. (**a**) Energy, (**b**) Carbohydrate, (**c**) Fat, and (**d**) Protein.

**Table 1 nutrients-14-03420-t001:** Participants’ characteristics at baseline (*n* = 27).

Variables	*n* (%) or Mean ± SE
Age (years)	37.1 ± 0.6
Marital status (married)	17.0 (63.0)
Current position (nursing technician)	14.0 (51.9)
Education level (complete or incomplete graduation)	16.0 (59.2)
Lifetime exposure to night work (years)	9.1 ± 0.7
The main reason to work at night (reconcile work with home and/or children’s care)	11.0 (40.7)
Second job (yes)	2.0 (7.4)
Smoking (no)	27.0 (100.0)
Alcohol intake (only on special occasions)	17.0 (63.0)
Physical activity (none)	17.0 (63.0)
BMI (kg/m^2^)	29.8 ± 0.4
CPD (hours)	2.9 ± 0.2
Chronotype (hours)	3.3 ± 0.2

BMI: body mass index; CPD: composite phase deviations.

**Table 2 nutrients-14-03420-t002:** Effect of melatonin and placebo administration on the quali/quantitative aspects of the diet.

Variables	Mean ± SE		Effects (*p*)	Goodness of Fit ***
% TEI *	Melatonin	Placebo	Intervention **	
Qualitative aspect				
Unprocessed or minimally processed foods	49.2 ± 3.2	52.6 ± 3.5	0.17	12.1
Processed foods	15.3 ± 1.9	13.1 ± 1.8	0.43	23.5
Ultra-processed foods	34.7 ± 2.4	34.8 ± 3.2	0.98	15.2
Quantitative aspect				
Carbohydrate	46 ± 9.7	44.6 ± 9.0	0.89	10.4
Protein	20.7 ± 10.2	21.3 ± 10.0	0.67	13.9
Fat	32.1 ± 7.9	33.1 ± 8.0	0.80	13.6

* Total energy intake. ** Means of the effect of melatonin and placebo administration, respectively, in association with CPD: unprocessed or minimally processed foods: 49.9 ± 9.1, 50.1 ± 9.8, *p* = 0.27; processed foods: 14.3 ± 5.0, 14.1 ± 5.2, *p* = 0.33; ultra-processed foods: 34.8 ± 4.9, 34.9 ± 8.1, *p* = 0.95; carbohydrate: 46.5 ± 10.9, 44.6 ± 10.5, *p* = 0.95; protein: 20.9 ± 12.2, 20.4 ± 10.7, *p* = 0.60; fat: 31.8 ± 10.7, 34.4 ± 11.8, *p* = 0.56. Means of the effect of melatonin and placebo administration, respectively, in association with chronotype: unprocessed or minimally processed foods: 49.1 ± 9.3, 52.6 ± 9.8, *p* = 0.22; processed foods: 15.4 ± 6.4, 13 ± 4.4, *p* = 0.62; ultra-processed foods: 34.6 ± 7, 34.9 ± 7.9, *p* = 0.14; carbohydrate: 46 ± 3.9, 44.7 ± 5, *p* = 0.56; protein: 20.7 ± 3.6, 21.3 ± 4.2, *p* = 0.48; fat: 32.1 ± 3.6, 33 ± 2.7, *p* = 0.81. *** Quasi-likelihood under independence model criterion (QIC) of interaction. All models are adjusted by lifetime exposure to night work (years).

## Data Availability

The data presented in this study are available on request from the corresponding authors. The data are not publicly available due to privacy and ethical reasons.

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
