# Peer review of "The Effect of Exogenous Melatonin on Eating Habits of Female Night Workers with Excessive Weight"

_nutrients, 2022, doi:10.3390/nu14163420_

Round 1

Reviewer 1 Report

This is an interesting and well-written  original article. I have only few minor comments:

- please provide a  figure with the plan of the study protocol

- results regarding Composite Phase Deviations should be presented in the table or fugure.

Sincerly yours

Author Response

We appreciate the opportunity to improve the quality of the manuscript “The effect of exogenous melatonin on eating habits of female night workers with excessive weight”. Point by point response to the comments of the reviewer follows below:

  • Please provide a figure with the plan of the study protocol.
    Thank you for your suggestion. The study flow chart has been provided as a supplementary material (Figure S1);

  • Results regarding Composite Phase Deviations should be presented in the table or figure.
    Thank you for your observation. Descriptive results on CPD are provided in Table 1, and the means of the effect of melatonin and placebo administration, respectively, in association with CPD are provided in Table 2.

Once again, we thank you for your valuable contributions and are available for any necessary clarifications.

Reviewer 2 Report

Thankyou for the opportunity to review this manuscript, the authors should be commended for a well-written and interesting manuscript. This manuscript presents novel findings on the eating habits of female night workers with excessive weight after melatonin administration. I have minor comments for the authors to consider to strengthen this manuscript: 

Introduction: 

- could more of a justification be provided for focusing on female participants

- line 68: the authors state that three studies have evaluated the effects of exogenous melatonin on 68 night workers in real-life conditions. Could some brief detail be provided of the results of these studies in order to provide context and direction for the reader. 

Discussion: 

- an additional consideration for understanding the lack of effect on meal timing could be the many factors that influence meal timing in shiftworkers. Perhaps the melatonin administration was working to impact satiety etc and had an impact on leptin/ghrelin, however because of a lack of control over meal timing at work etc then workers were unable to change when they ate. Leading to the lack of difference in the findings. 

Author Response

We appreciate the opportunity to improve the quality of the manuscript “The effect of exogenous melatonin on eating habits of female night workers with excessive weight”. Point by point response to the comments of the reviewer follows below:

  • Could more of a justification be provided for focusing on female participants.
    Thank you for your suggestion. Rationale has been provided in line 81 as follows: “The choice of focusing on females is due to the higher prevalence of excessive weight in women [27], and because nursing is a female-dominated occupation [28].”;

  • Line 68: the authors state that three studies have evaluated the effects of exogenous melatonin on night workers in real-life conditions. Could some brief detail be provided of the results of these studies in order to provide context and direction for the reader.
    Thank you, we appreciate your suggestion. Detail has been provided in line 69 as follows: “These studies assessed the effect of melatonin on 1) circadian misalignment: administration prior to daytime sleep attenuated interference from circadian alerting process, demonstrating both phase-shifting and sleep-promoting actions [22]; 2) sleep and alertness: administration improved sleep problems and increased alertness during working hours [23], and 3) body weight: administration reduced body weight, BMI, waist and hip circumferences [24].”;

Once again, we thank you for your valuable contributions and are available for any necessary clarifications.